



# Assimilation of ice compactness data in a strong coupling regime in the ocean - sea ice coupled model

Maxim N. Kaurkin[1], Leonid Y. Kalnitski[2,1], Konstantin V. Ushakov[1,3], and Rashit A. Ibrayev[2,1,3]

[1]Shirshov Institute of Oceanology, Russian Academy of Sciences, Nahimovskiy prospekt, 36, Moscow, 117997, Russia
[2]Marchuk Institute of Numerical Mathematics, Russian Academy of Sciences, ul. Gubkina, 8, Moscow, 119333, Russia
[3]Moscow Institute of Physics and Technology (State University), Institutskiy per. 9, Dolgoprudny, Moscow reg., 141700, Russia

*Correspondence to:* Maxim N. Kaurkin (maksim.kaurkin@phystech.edu)

**Abstract.**

The Arctic Ocean plays an important role in the global climate system, where sea ice regulates the exchange of heat, moisture and momentum between the atmosphere and the ocean. A comprehensive analysis and forecast of the Arctic ocean system requires a detailed numerical ocean and sea ice coupled model supplemented by assimilation of observational data at
appropriate time scales.

A new operative ocean – ice state forecast system was developed and implemented. It consists of the INMIO4.1 ocean general circulation model and the CICE5.1 sea ice dynamics and thermodynamics model with common spatial resolution of $0.25°$. For the exchange of boundary conditions and service actions (data storage, time synchronization, etc.), the coupled model uses the Compact Modeling Framework (CMF3.0). Data assimilation is implemented in the form of the Data Assimilation Service
(DAS) based on the Ensemble Optimal Interpolation (EnOI) method. This technique allows to simultaneously correct the ocean (temperature, salinity, surface level) and ice (concentration) model fields in the DAS service, so they are coordinated not only through the exchange of boundary conditions, but already at the stage of data assimilation (i.e. strong coupling data assimilation). Experiments with the INMIO - CICE model show that the developed algorithm provides a significant improvement in the accuracy of forecasting the state of the ice field in the Arctic Ocean.

## 15   1   Introduction

The Arctic Ocean plays an important role in the global climate system, where sea ice regulates the exchange of heat, moisture and momentum between the atmosphere and the ocean. The warming of the Arctic region and shifts in its water cycle observed in the last decades are accompanied by a wide range of processes in the ice-ocean system, see review (Guemas et al., 2016).

Interpretation of such changes is very difficult due to the lack of observational data. Numerical modeling can greatly con-
tribute to understanding these processes, but the lack of knowledge about the physics of ice-ocean interactions limits our ability to realistically reproduce them. The straightforward remedy is to correct the dynamic model solution by assimilating the observations available at the appropriate time scales.





The practice of numerical weather prediction has shown that coupled models usually provide significantly more accurate forecasts than stand-alone ones. This has led to research on data assimilation (DA) in coupled models, which promises to further improve forecast quality (Skachko et al., 2019). An overview of current activities in coupled forecasting systems and coupled DA can be found in (Brassington et al., 2015). The World Meteorological Organization (WMO) Meeting on Coupled

Data Assimilation (Penny and Hamill, 2017) defined the classification of weakly coupled and strongly coupled DA and their variations.

The forecasting experience of leading scientific centers has shown that for an accurate mid-term forecast of the state of water and ice in the Arctic, the models of ocean and ice dynamics and thermodynamics need to be fully coupled. Thus, a DA technique capable of retaining sea ice in a coupled model in a dynamically consistent manner could help improving the

accuracy of climate studies and weather predictions (Kimmritz et al., 2018). The goal of this work is to develop a coupled ocean – sea ice model with coupled assimilation of available observational data and implement it with application for the Arctic region.

## 2    Coupled ocean-ice model with data assimilation

### 2.1    Coupled ocean-ice model setup

In this work, we use the global ocean – sea ice coupled model, which consists of the INMIO ocean general circulation model (Ibrayev et al., 2012) and the CICE ice dynamics and thermodynamics model (Hunke et al., 2015) operating on massively parallel computers under control of the Compact Modeling Framework (CMF) (Kalmykov et al., 2018).

The ocean and sea-ice models use similar three-polar grids (Murray, 1996) with $0.25°$ nominal resolution. The number of ocean z-grid horizons is 49 with vertical steps increasing from 6 m near the surface to 250 m in the deep part. The bottom

topography data were interpolated from the (ETOPO5) array, excluding inland water bodies and small islands.

The INMIO-CICE model setup is the same as used in the ocean-ice component of (Fadeev et al., 2019). In particular, for the ocean model the lateral momentum exchange is modelled by the biharmonic operator with the coefficient equal to $-1.5 \cdot 10^{11} m^4/s$ on the equator and scaled proportionally to the grid cell area in power $3/2$. The additional Smagorinsky biharmonic term in the form of (Griffies and Hallberg, 2000) is added to ensure numerical stability. The lateral heat and salt

mixing is approximated by the Laplacian operator with coefficient $300 m^2/s$ on the equator and scaled proportionally to the grid cell area in power $1/2$. The momentum advection is approximated by the leap-frog scheme, while for the heat and salt advection the flux-corrected transport scheme (Zalesak, 1979) is used. The atmosphere-ocean surface fluxes of heat, mass and momentum are calculated by the (Launiainen and Vihma, 1990) atmospheric boundary layer bulk-formulae.

In the ice model configuration, the ice in each grid cell is split into five thickness categories with one additional category for

snow. Thus, the array of ice concentrations (i.e. fractions of the grid cell area) for each category is used as the main variable characterizing the state of the ice in the model. The elastic-viscous-plastic approximation is used to model the sea ice rheology, the ice transport is performed with the upwind scheme. To describe the evolution of temperature, a zero-layer thermodynamic model is used, in which ice is considered fresh and has zero heat capacity. A similar configuration of the CICE model is used





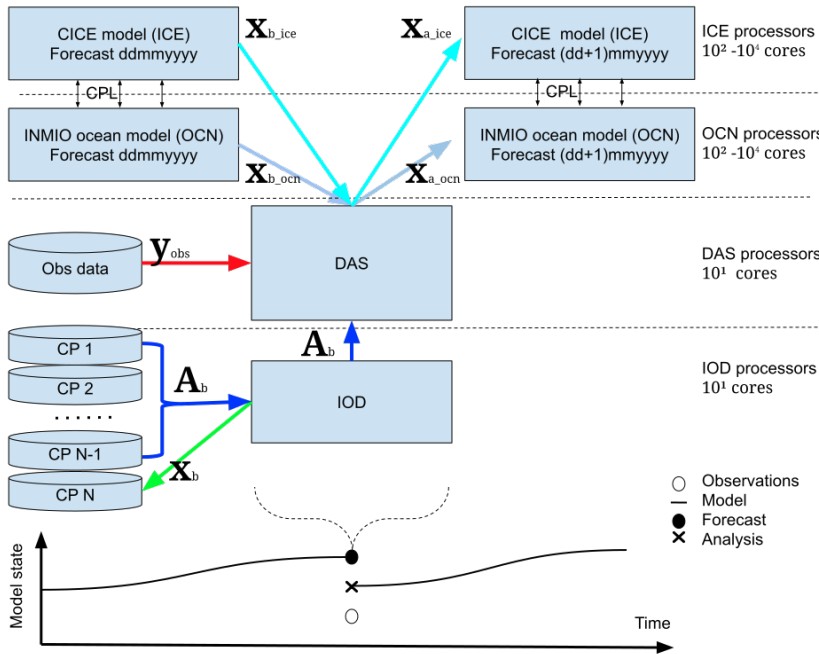

**Figure 1.** Schematic diagram of data transfer between the ocean model (OCN), the sea ice model (ICE), the Data Assimilation Service (DAS), and the I/O service (IOD) within the coupled model under the CMF framework. The horizontal dotted lines separate the CPU cores allocated to the models and services for the case of 0.25° global resolution.

in many key centers for operational forecasting of the ocean state, in particular in the TOPAZ system (Norway) consisting of a coupled ocean-sea ice model (HYCOM / CICE) with DA by the EnKF method (ensemble Kalman filter) (Sakov et al., 2012).

The time step for both INMIO and CICE components is 10 minutes, while their coupling interval is 20 minutes. The fields of upper grid layer temperature, salinity, velocity vector, freezing/melting potential, and surface tilt vector are sent from the
ocean to the ice model. The fields (intergral over all categories) of ice concentration, horizontal ice-water stress vector, fluxes of fresh water, salt, sensible heat and penetrating short-wave radiation are sent from the ice to the ocean model.

Before the experiments on the assimilation of observational data, the model spin-up was performed for the period of 2009.01.01 – 2019.08.31, during which the atmospheric near-surface fields were defined by the ERA-Interim reanalysis (Dee et al., 2011) and the model solution was saved every 10 days. Further, these states, starting from the year 2011, were used as
elements of the ensemble for the approximation of covariance matrices during the DA experiment (see Section 3).

Figures 3 and 4 show the fields of sea surface temperature and ice concentration in the *h01* and *h02* experiments in comparison with the independent OSTIA (Roberts-Jones et al., 2012) observational data for February 1, May 1, July 1, September 1, and December 1 2020. It can be seen from the figures that the model fields without assimilation are excessively smooth. No eddy dynamics is observed, which is typical for models with a coarse resolution. It is also worth noting that an overestimated
amount of sea ice is produced by the coupled model, which is successfully corrected by assimilation. Also due to the assim-





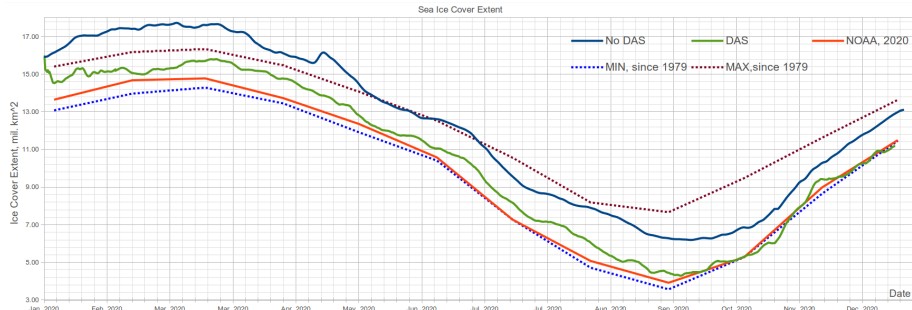

**Figure 2.** National Snow and Ice Data Center (NSIDC) sea ice extent data and INMIO-CICE model simulations results with and without data assimilation via DAS in 2020 for the Northern Hemisphere. The dotted lines show the minimum and the maximum according to the NSIDC data for 1979-2019.

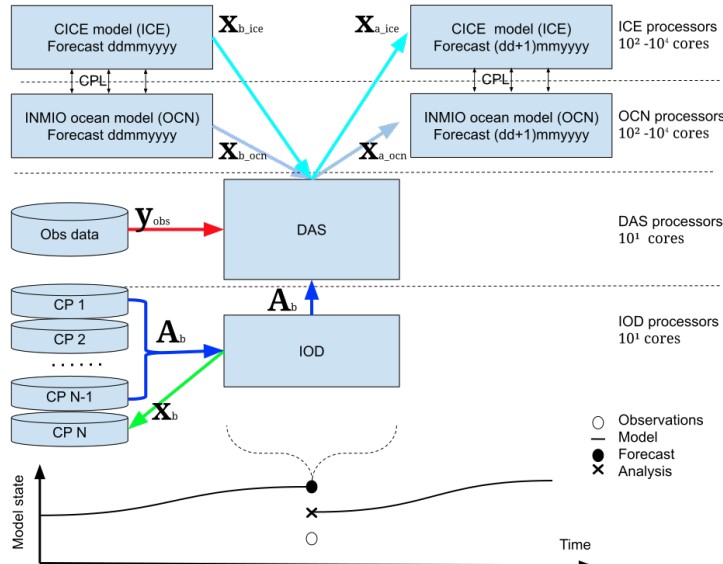

**Figure 3.** Sea surface temperature fields in the *h01* (no DAS) and *h02* (DAS) experiments in comparison with the OSTIA observational data for February 1, May 1, July 1, September 1, and December 1 2020. The red dots show tracks of assimilated satellite sea surface level data, the triangles show locations of the Argo temperature and salinity profiles. The white crosses show the locations of assimilated data on the concentration of sea ice, which are within the limits of $[0.1, 0.9]$, that is, the boundary of the ice cover and the area where there is active melting.

ilation, the surface temperature field becomes more gradient, and the area of the ice cover decreases significantly, so that the model solution becomes more consistent with the OSTIA data.





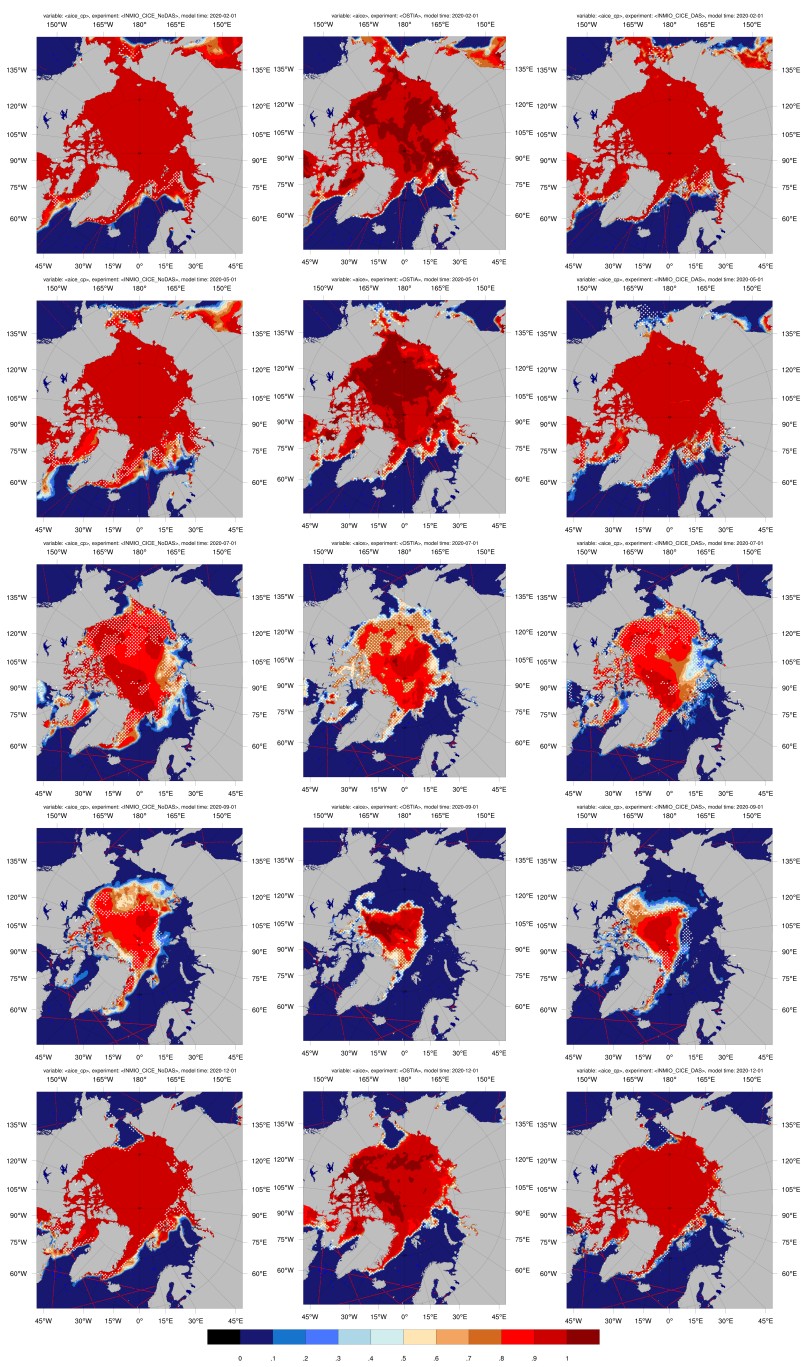

**Figure 4.** Sea ice concentration fields in the *h01* (no DAS) and *h02* (DAS) experiments in comparison with the OSTIA data for February 1, May 1, July 1, September 1, and December 1 2020.


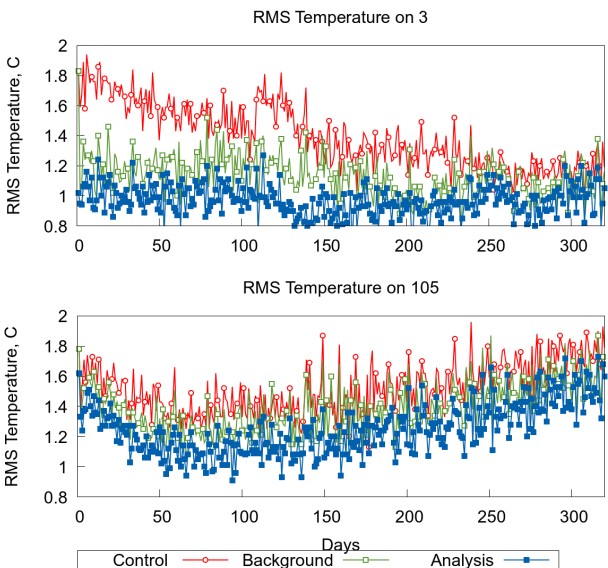

**Figure 5.** The RMS error of model temperature (°C) with respect to the data of Argo floats for experiments *h01* (Control), *h02* forecast (Background) and *h02* Analysis at depths of 3 m and 105 m.

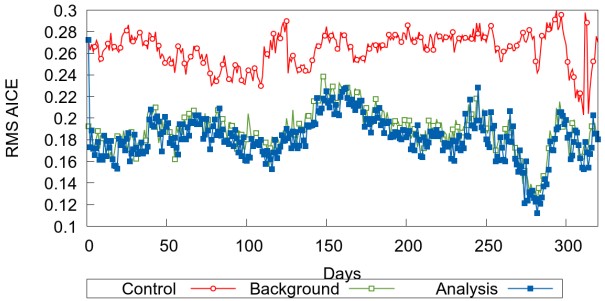

**Figure 6.** The RMS error of the sea ice concentration (area fraction) with respect to the OSI SAF data for experiments *h01* (Control), *h02* forecast (Background) and *h02* Analysis.

## 2.2 EnOI Data Assimilation

The basic equations of the Ensemble Optimal Interpolation (EnOI) method are as follows (Evensen, 2003):

$$x_a = x_b + \mathbf{K}(y_{obs} - \mathbf{H}x_b) \qquad\qquad \mathbf{K} = \mathbf{B}\mathbf{H}^T(\mathbf{H}\mathbf{B}\mathbf{H}^T + \mathbf{R})^{-1} \qquad\qquad (1)$$

where $x_b$ (background) and $x_a$ (analysis) are the vectors of size *n* representing the model solution before and after DA, respectively;



$n$ is the number of model grid points weighted by the number of model variables to be corrected (temperature, salinity, sea

level, etc.);

$\boldsymbol{y_{obs}}$ is the vector of observations of size $m$;

$m$ is the total number of observation points where various data were obtained;

$\mathbf{K}(n \times m)$ is the gain matrix;

$\mathbf{R}(m \times m)$ is the covariance matrix of observation errors;

$\mathbf{H}(m \times n)$ is the matrix, representing the projection operator of model values into the observational data space;

$\mathbf{B}$ is the the covariance matrix of model errors.

The basic idea of ensemble methods (EnKF and EnOI) is that the matrix $\mathbf{B}$ is approximated on the basis of a set (ensemble) of model solution vectors (Evensen, 2006). Let's define the ensemble matrix $\mathbf{A}_b^N = [\boldsymbol{x}_b^1 \dots \boldsymbol{x}_b^N] - [\overline{\boldsymbol{x}_b} \dots \overline{\boldsymbol{x}_b}]$ of size $(n \times N)$, where $N$ is the ensemble size, and the matrix columns are equal to the model states minus their average over the ensemble.

Then the covariance matrix of model errors can be approximated on the basis of this ensemble in the following way:

$$\mathbf{B} \approx \mathbf{B}^N = \frac{1}{N-1} \mathbf{A}_b^N (\mathbf{A}_b^N)^T \qquad (2)$$

## 2.3   Program implementation of EnOI

The assimilation procedure is encapsulated in the Data Assimilation Service (DAS) of the CMF framework. The service works

in parallel on its separate CPU cores and can be used simultaneously for several model components (ocean, ice, atmosphere, etc.). At the same time, there is no explicit dependence on the model equations, difference schemes, and parameterizations. Only the model output is used in the form of ensemble vectors, on the basis of which the covariance matrix $\mathbf{B}$ is approximated. The model forecast $\boldsymbol{x_b}$ is supplied to the service regularly (e.g., once per model day) from all model components via the cluster interconnect without accessing the file system.

The schematic diagram of data transfer between the ocean model (OCN), the sea ice model (ICE), the Data Assimilation Service (DAS), and the I/O service (Input-Output Data, IOD) within the coupled model under the CMF framework is shown on Figure 1. The figure also shows estimates of the required number of CPU cores to run the global model with a resolution of $0.25°$. Parallel scaling issues of the INMIO-CMF model were discussed in (Kalmykov et al., 2018).

To approximate the model error covariance matrix $\mathbf{B}$ (Kaurkin et al., 2016a), we used the ensemble of the ocean and sea ice

model states. Namely, each element of the ensemble includes the 3D-arrays of temperature $T(x,y,z)$ and salinity $S(x,y,z)$, and the 2D-arrays of sea surface level anomaly $SSL(x,y)$ and sea ice concentration $AICE(x,y)$, merged together into a 1D-stripe (see Section 3). Columns of the matrix $\mathbf{B}$ consist of these stripes minus their ensemble mean.

The ensemble size was chosen equal to 50 based on the error decreasing rate estimated in the numerical experiments (Kaurkin et al., 2016b), (Kaurkin et al., 2016a). The covariance matrix $\mathbf{B}$ calculated in this way will take into account the correlation

(relationship) between different model variables: sea level, ice concentration, temperature and salinity at different model horizons, see Table 1. Thus, assimilation of even one of the variables will correct the entire vector of model solution.





**Table 1.** Contents and sizes of the vectors $x_a, x_b, y_{obs}$, and the matrix $\mathbf{A}_b^N$. $T$, $S$ profiles data from the Argo project; absolute dynamic topography (ADT) along-track data from the Jason-3 satellite, AVISO project; sea ice concentration (SIC) data from the EUMETSAT OSI SAF project.

| Element | Model state vectors (forecast $x_b$ and analysis $x_a$) | Observation data vector $y_{obs}$ | Ensemble matrix $\mathbf{A}_b$ |
|---|---|---|---|
| Size | $n \sim 10^8$ | $m \sim 10^5$ | $n \times N \sim 10^{10}$ |
| Contents | ocean $T$ (x, y, z) | $T$ (Argo) | model state vector |
| | ocean $S$ (x, y, z) | $S$ (Argo) | $(T, S, SSL, AICE)$ |
| | ocean $SSL$ (x, y) | $ADT$ (AVISO, Jason-3) | saved every 10 days |
| | ice $AICE$ (x, y) | $SIC$ (OSI SAF) | for 2011-2019 |

In DAS, the seasonally varying ensemble (Oke et al., 2010) is implemented. For each call of the assimilation service, the model states for the same calendar month, but for previous years of model run, are used for the production of the ensemble. Since only anomalies of the ensemble elements are used in the assimilation procedure, this approach allows to eliminate
seasonal covariances, while preserving the mutual influence of synoptic circulation elements.

The following observational data will be used:

- In situ profiles of temperature and salinity from Argo floats (Argo, 2021)

- Absolute dynamic topography (ADT) data from the AVISO project (Jason-3 satellites) (Bannoura et al., 2013)

- The AMSR-2 sea ice concentration (indicates the fraction of a given ocean grid point covered by ice) product of the
EUMETSAT Ocean and Sea Ice Satellite Application Facility (OSI SAF) (Tonboe et al., 2017)

## 2.4 Features of data assimilation in the ocean-ice model

The widely used weakly coupled DA for the ocean-ice coupled model, in the form of nudging DA method for the CICE model and EnOI method for the INMIO model, proved to be inappropriate for our tasks. That is, the integration of the ocean-ice model with this DA scheme regularly led to inconsistencies of the ocean and ice fields. As a consequence, the coupled model
5 crashed due to the violation of the dynamic-thermodynamic balance in it, getting physically incorrect model fields and an *"ice thermo error"* or *"wrong ice energy error"* or *"ridging error"* in the CICE model.

This indicates that there is a complex dynamic-thermodynamic balance in CICE, when a sharp change in one of the fields within the ice model or the arrival of inconsistent boundary conditions from the ocean model can lead to errors arising in the calculation of other fields. Thus, it is necessary to coordinate the thermodynamic and dynamic fields at the stage of their
10 correction by DA, in particular, to coordinate the temperature and salinity fields in the ocean and the ice concentration and thickness fields in the ice model.



So, the strongly coupled data assimilation approach was implemented in the ocean-ice model, when DA for different components is executed in the separate DAS service with coupled analysing fields from the two models. However, for smooth mutual adjustment, it was nevertheless required to implement an interface for the ice model, in which the correction factor field for ice concentration ($K_{aice}$) is introduced. The correction factor is equal to the model analysis obtained by the EnOI algorithm ($x_a$) divided by the background ($x_b$), and is restricted to the range $[1/\alpha, \alpha]$, where $\alpha$ is an empirical parameter. In our experiments, the optimal value of $\alpha$ is 4. At grid nodes where correction is not required or the model background has no ice the $K_{aice}$ coefficient is set to 1. Further, to correct the calculations in the CICE model, ice concentrations of various thickness categories (five ice categories and one snow category) are multiplied by the $K_{aice}$ field and the results are renormalized so that the integral concentration field is less than 1 at all nodes.

Thus, due to DA, there is a gradual "freezing" or "melting" of ice without direct correction of the ice thickness field. The advantage over weakly coupled DA is that during the consequent correction of the ice concentration, other model fields (temperature, salinity, sea level) are also taken into account, thus they are coordinated not only through the mechanism of exchange of boundary conditions, but already at the stage of DA.

## 3  Experiments on simulation of water and sea ice circulation in the Arctic in 2020.

The proposed algorithm for data assimilation in a strongly coupled mode was tested in a series of numerical experiments with the global INMIO - CICE coupled ocean – ice model with 0.25°resolution. The model was forced with atmospheric data of the (GFS) forecasts accumulated for the period of 2020.01.01 – 2020.12.31. Two experiments were carried out:

1. *h01* - the control run without assimilation of observation data;

2. *h02* - the experiment with strongly coupled assimilation of observed temperature, salinity, ocean surface level and sea ice concentration.

In this article, we will focus on the reproduction of the characteristics of sea ice only. Details of reproducing other characteristics of ocean circulation will be considered in separate articles.

### 3.1  Ice extent in the Arctic in 2020

Figure 2 and Table 2 compare the annaul variations of Northern Hemisphere sea ice extent in 2020 obtained in experiments *h01* (no DAS) and *h02* (DAS) with analysis data provided by the National Snow and Ice Data Center (NSIDC). It can be seen that due to the assimilation of observations, it was possible to significantly improve the accuracy of sea ice cover simulation. The average of monthly errors decreased from 26 % to 6%. Figure 2 also shows that in the no-assimilation experiment *h01*, the resulting ice extent did not even fit into the "corridor" of interannual variability for 1979-2019. On the other hand, with assimilation in *h02* we can see a clear correspondence of the simulation with the analysis data.





**Table 2.** Comparison of the year 2020 Northern Hemisphere sea ice extent data provided by the National Snow and Ice Data Center (NSIDC) with INMIO-CICE simulations without DA (*h01*) and with DA (*h02*).

| 2020 month | $h01$, $m.km^2$ | $\frac{h01-NSIDC}{NSIDC}$,% | $h02$, $m.km^2$ | $\frac{h02-NSIDC}{NSIDC}$,% | NSIDC, $m.km^2$ |
|---|---|---|---|---|---|
| Jan | 16.63 | 22 | 14.97 | 10 | 13.65 |
| Feb | 17.45 | 19 | 15.17 | 3 | 14.68 |
| Mar | 17.50 | 18 | 15.58 | 5 | 14.78 |
| Apr | 16.13 | 17 | 14.66 | 7 | 13.73 |
| May | 14.44 | 17 | 12.73 | 3 | 12.36 |
| Jun | 12.49 | 18 | 11.02 | 4 | 10.58 |
| Jul | 9.62 | 32 | 8.09 | 11 | 7.28 |
| Aug | 7.77 | 53 | 5.93 | 17 | 5.08 |
| Sep | 6.42 | 64 | 4.56 | 16 | 3.92 |
| Oct | 7.03 | 33 | 5.55 | 5 | 5.28 |
| Nov | 10.28 | 14 | 9.05 | 1 | 8.99 |
| Dec | 13.32 | 13 | 11.86 | 1 | 11.77 |

## 3.2 Forecast error

Figures 5 and 6 show the root-mean-square (RMS) error plots of the temperature and ice concentration fields north of the $30°N$ latitude for the control experiment with no assimilation *h01* and for the experiment *h02* before assimilation (background) and after it (analysis). From these graphs we can conclude that assimilation gives the correct sign of the correction, and the difference between the RMS errors in the assimilating experiment and the control one is $\sim 0.5°C$. Furthermore, the absolute magnitude of the RMS error over all observation points for the SST is $\sim 1°C$, and $\sim 0.2$ for the ice concentration. This is in agreement with data from the Copernicus Marine Environment Monitoring Service (CMEMS) (Melsom et al., 2019), in which the RMS error of the daily forecast is $\sim 0.8°C$ for SST , and $0.2$ for the sea ice concentration. The methodology for calculating errors of sea ice concentration simulations is presented in (Desportes et al., 2017), (Melsom et al., 2019).

## 4 Conclusions

A strongly coupled data assimilation scheme was proposed and tested for the ocean - sea ice model INMIO-CICE with a wide range of observational data (Argo temperature and salinity, AVISO absolute dynamic topography of the sea level, OSISAF sea ice concentration). The main goal of the numerical experiments was to check the stable operation of the coupled ice-ocean prediction system, the correctness of the resulting model fields and the suitability of the entire system for operational calculations.





A retrospective calculation was performed to reproduce the sea ice cover and surface temperature fields for the year 2020. A comparison was made with independent observational data on the SST and ice concentration fields (OSTIA) and the sea ice extent provided by the National Snow and Ice Data Center (NSIDC) for the Northern Hemisphere, which showed good

10 performance of the proposed method.

*Code availability.* The code of the INMIO-CICE-CMF (distributed under GPLv2 licence) is available on http://model.ocean.ru

*Data availability.* The INMIO-CICE-CMF3.0-DAS simulation results for 2020 are available at the link:
http://ivm-nat.nc.mstn.ru:40223/INMIO_CICE_DAS_OS

*Acknowledgements.* The research was supported by the Russian Science Foundation (project no. 19-77-00104) and performed at the Shirshov Institute of Oceanology, Russian Academy of Sciences. The calculations were carried out using the supercomputer resources of the Joint Supercomputer Center, Russian Academy of Sciences (www.jscc.ru) and the SUGON cluster of the Marine Hydrophysical Institute, Russian Academy of Sciences.

5 *Author contributions.* MK was the author of the Data Assimilation Service (design, code writing, and testing), carried out numerical experimants and wrote the first draft of the article. LK was responseble for operation the sea ice model CICE. RI and KU designed and developed the INMIO model and physical coupling algorithms.



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
