# Peer review of "Assimilation of ice compactness data in a strong coupling regime in the ocean - sea ice coupled model"

_Ocean Science, 2021_

## Referee Comment (RC1)

Referee comment – Assimilation of ice compactness data in a strong coupling regime in the ocean – sea ice coupled model

General comments:
In general, the manuscript is well written and easy to read. The manuscript describes the model INMIO-CICE model system well and shows that with assimilation the sea-ice concentration forecast is improved. However, I struggle a bit to see the novelty of this work. In the last 10-20 years there has been a lot of papers of coupled ocean sea-ice models with assimilation and they have all shown improvements in sea-ice concentration (Caya et al. 2010, Sakov et al. 2012, Buehner et al. 2013, Yang et al. 2014, Posey et al. 2015, Xie et al. 2016, Mu et al. 2018, Fritzner et al. 2019,2020). I understand that the model system used here is probably new, so to get a grasp of the novelty I think the manuscript would severely benefit from a description of why this model system is different from the others than have been used before and which improvements that can be expected.

Scientific comments:
I think the results section is lacking for the evaluation of a new assimilation system. Both ocean and ice observations are assimilated and thus it is natural that both ice and ocean output in the model is evaluated to see what effect the observations have. In general, sea-ice concentration evaluation with a 1-day assimilation time step on a 0.25 degrees grid is not so interesting because the day to day variations on these scales are so small anyways. It would also be interesting with a verification of a variable not observed such as sea-ice thickness to see if there is any update on this variable based on the estimated error covariance matrix, you could for example use the SMOS observations (https://earth.esa.int/eogateway/catalog/smos-l3-sea-ice-thickness). I think also that the sea-ice concentration verification could benefit from using some sea-ice edge distance metrics (Melson 2019) in addition to RMS error. In the results/conclusion section you should also discuss the results in relation to other studies that have done similar assimilation experiments.

Specific comments:
p.3 l.10-15: It's a bit strange to have the results described here in the setup section before you have described the assimilation at all.

Figure 3.: This figure is wrong.

Figure 4.: Please use some headings on the columns and rows (or letters) so it is easier to understand which experiments we are looking at and which dates are which. Why do I see these strange dots in the marginal ice zone? Is it because of low resolution in the images?

Figure 5.: Add the dimension in the titlte: E.g. RMS Temperature on 3m (depth also maybe)

p.7 l.16: Also include the dimension of B (n x n)

p.9 l.15-16: Why xa over xb?

p.9 l.17: How do you know when a correction is required?

p.9 l.12-19: Please describe the use of correction factor further, i.e. how it is implemented and why it is needed.

p.9 l.21-24: I don't understand, please describe further.

p.9 l.10-11: See comments above, I don't think this is enough to evaluate the model system.

References:

Caya, Alain, Mark Buehner, and Tom Carrieres. "Analysis and forecasting of sea ice conditions with three-dimensional variational data assimilation and a coupled ice–ocean model." *Journal of Atmospheric and Oceanic Technology* 27.2 (2010): 353-369.

Sakov, Pavel, et al. "TOPAZ4: an ocean-sea ice data assimilation system for the North Atlantic and Arctic." *Ocean Science* 8.4 (2012): 633-656.

Buehner, Mark, et al. "A new Environment Canada regional ice analysis system." *Atmosphere-Ocean* 51.1 (2013): 18-34.

Yang, Qinghua, et al. "Assimilating SMOS sea ice thickness into a coupled ice-ocean model using a local SEIK filter." *Journal of Geophysical Research: Oceans* 119.10 (2014): 6680-6692.

Posey, Pamela G., et al. "Improving Arctic sea ice edge forecasts by assimilating high horizontal resolution sea ice concentration data into the US Navy's ice forecast systems." *The Cryosphere* 9.4 (2015): 1735-1745.

Xie, Jiping, et al. "Benefits of assimilating thin sea ice thickness from SMOS into the TOPAZ system." *The Cryosphere* 10.6 (2016): 2745-2761.

Mu, Longjiang, et al. "Improving sea ice thickness estimates by assimilating CryoSat-2 and SMOS sea ice thickness data simultaneously." *Quarterly Journal of the Royal Meteorological Society* 144.711 (2018): 529-538.

Fritzner, Sindre, et al. "Impact of assimilating sea ice concentration, sea ice thickness and snow depth in a coupled ocean–sea ice modelling system." *The Cryosphere* 13.2 (2019): 491-509.

Fritzner, Sindre, Rune Graversen, and Kai H. Christensen. "Assessment of High-Resolution Dynamical and Machine Learning Models for Prediction of Sea Ice Concentration in a Regional Application." *Journal of Geophysical Research: Oceans* 125.11 (2020): e2020JC016277.

Melsom, Arne, Cyril Palerme, and Malte Müller. "Validation metrics for ice edge position forecasts." *Ocean Science* 15.3 (2019): 615-630.